

# Evaluation of factors affecting TOC and its trend at three Antarctic stations in the years 2007–2023

David Tichopád[1], Kamil Láska[1], Tove Svendby[3], Klára Čížková[1,2], Andrea Pazmiño[4], Boyan Petkov[5,6], Ladislav Metelka[2]

[1]Masaryk University, Faculty of Science, Department of Geography, Brno, Czech Republic
[2]Solar and Ozone Observatory, Czech Hydrometeorological Institute, Hradec Králové, Czech Republic
[3]The Climate and Environmental Research Institute NILU, Kjeller, Norway
[4]LATMOS/IPSL, UVSQ, Université Paris-Saclay, Sorbonne Université, CNRS, Guyancourt, France
[5]Department of Advanced Technologies in Medicine & Dentistry, University G. d'Annunzio, Chieti-Pescara, Italy
[6]Institute of Polar Sciences, National Research Council, Bologna, Italy

*Correspondence to*: David Tichopád (david.tichopad@mail.muni.cz)

**Abstract.** This study assesses trends in the total ozone column (TOC) and the atmospheric factors influencing ozone variability at three Antarctic stations (Marambio, Troll/Trollhaugen, and Concordia) from 2007 to 2023. Ground-based TOC measurements were used, supplemented by satellite observations from the Ozone Monitoring Instrument on NASA's Aura satellite. TOC trends were derived using a multiple linear regression model provided by the Long-term Ozone Trends and Uncertainties in the Stratosphere (LOTUS) project. The selected LOTUS model was able to explain 94–97 % of the TOC variability at all three stations. The regression analysis showed that ozone variability at these stations is mainly driven by the lower stratospheric temperature, eddy heat flux, and the Quasi-Biennial Oscillation. A statistically significant increasing trend was found at the Marambio station (3.43 DU/decade), while statistically insignificant trends were detected at the other two stations. Using MERRA-2 reanalyses, the LOTUS model was applied to each grid point in the 40–90° S region, which effectively illustrates the spatial distribution of the impacts of individual predictors. It was found that warmer conditions in the Antarctic stratosphere in September 2019 caused TOC to be up to 100 DU higher than normal, especially over East Antarctica. The results improve understanding of regional TOC trends and how the Antarctic ozone layer responds to changes in ozone-depleting substances.

## 1 Introduction

The stratospheric ozone layer is very important for life on Earth because it protects the biosphere from the harmful solar UV radiation (Brasseur and Solomon, 2005). The discovery of the Antarctic ozone hole (Chubachi, 1984; Farman et al., 1985; Solomon et al., 1986) has been important, demonstrating the need to protect the ozone layer and leading to the Vienna Convention in 1985 and the signing of the Montreal Protocol in 1987. Due to this successful international treaty, the emission of ozone-depleting substances (ODS), which are the sources of ozone-depleting active halogens in the stratosphere, began to decline (e.g., Solomon, 1999). In response to the reduction of ODS in the stratosphere, the total ozone column (TOC) is





expected to recover globally. The ozone depletion is most severe at southern high latitudes, but in recent years, several studies (e.g., Solomon et al., 2016; Kuttippurath and Nair, 2017; Pazmiño et al., 2018; Weber et al., 2022) point out the possible onset of TOC recovery in Antarctic spring, especially in September. Despite the ongoing reduction in ODS concentrations, the

magnitude of chemical ozone depletion over Antarctica can fluctuate significantly from year to year, depending on varying meteorological and dynamical conditions (Newman et al., 2006; Keeble et al., 2014; de Laat et al., 2017; Tully et al., 2019; Stone et al., 2021).

The strong, cold polar vortex is the main feature of the winter stratospheric circulation over Antarctica, which greatly affects the severity of the annual ozone depletion. The presence of the polar vortex causes the isolation of the Antarctic stratosphere

from the surrounding air masses, creating unique chemical and dynamic conditions (Nash et al., 1996). Annually, the southern polar vortex forms during autumn, intensifies to its peak in mid-winter, and typically dissipates by November or December (Waugh and Polvani, 2010). Nevertheless, there have been instances of premature disruption; for example, in September 2019, a strong wave with wave number 1 developed in the southern stratosphere, triggering a sudden stratospheric warming (SSW) and an earlier breakdown of the polar vortex (Hendon et al., 2019; Eswaraiah et al., 2020; Yamazaki et al., 2020). Many

studies, such as Schoeberl et al. (1996), Newman and Nash (2000), and Newman et al. (2006), stress the importance of the polar vortex properties in relation to the variability of the Antarctic ozone loss. Additional influences on the Antarctic stratosphere and ozone loss may come from events such as wildfires or volcanic eruptions. Known cases are, for example, the extensive fires observed in Australia at the turn of 2019 and 2020 (Salawitch and McBride, 2022) and the strong eruption of the Hunga Tonga-Hunga Ha'api volcano in mid-January 2022 (Fleming et al., 2024; Kozubek et al., 2024).

Antarctic ozone trends have been extensively studied recently (e.g. Weber et al., 2018, 2022; Pazmiño et al., 2023; Jonson et al., 2023; Fioletov et al., 2023). However, less attention has been paid so far to the regional variability of TOC and the atmospheric factors that influence ozone variability in the southern polar regions.

The purpose of this study is to assess the 2007–2023 trends in Antarctic TOC and the factors affecting it, using the novel Long-term Ozone Trends and Uncertainties in the Stratosphere (LOTUS) regression method applied to ground-based data from three

Antarctic stations (Section 2). The set of regression predictors suitable for the southern high latitudes is defined in Section 3. Moreover, this study is the first to perform LOTUS regressions for Antarctic stations and for each grid point in the 40–90° S region using MERRA-2 reanalyses (section 4), leading to a better understanding of how the spatial patterns of total ozone column are affected by individual atmospheric predictors.





Figure 1: Location of the three Antarctic stations used in this study.

## 2 Data

### 2.1 Ground-based TOC data

This study analysed ground-based total ozone column (TOC) measurements from three Antarctic stations: Marambio, Troll & Trollhaugen, and Concordia. Marambio station (64.14° S, 56.37° E, 196 m a.s.l.) is a permanent Argentine base established in



1969, situated on the ice-free Seymour Island within the Antarctic Peninsula region (Fig. 1). Daily TOC observations at Marambio were obtained using the B199 double-monochromator Brewer spectrophotometer over the period from February 2010 to January 2020, during which the instrument was operated jointly by the Czech Hydrometeorological Institute (CHMI) and the National Meteorological Service of Argentina (NMSI). While the B199 instrument offers very high accuracy, reaching up to 0.15 % (Scarnato et al., 2010), only direct sun measurements were utilised. Calibration was performed regularly in

accordance with international standards, following the procedures described in Čížková et al. (2023).

Furthermore, daily TOC from the Troll station was used. Troll is Norway's only year-round research station in Antarctica, located in Jutulsessen, 235 km offshore in the eastern part of the Princess Martha Coast in Queen Maud Land, Antarctica. At the Troll station, data were obtained from the NILU-UV radiometer, which is a ground-based filter instrument with five UV channels at wavelengths centred around 302, 312, 320, 340, and 380 nm (Sztipa et al., 2019). From 27 January 2007 to 19

January 2014, the NILU-UV radiometer, designated 015, was located at 72.01° S, 2.535° E (1270 m a.s.l.). On January 30, 2014, the instrument was moved to a new location at Trollhaugen, approximately 1 km from Troll and 1553 m a.s.l. However, in January 2015, technical problems with the filters in the NILU-UV led to a decrease in data reliability, and measurements with the instrument 015 were terminated on 11 May 2015. A new instrument (number 005) was installed on 24 November 2015 at the Trollhaugen site and has been in continuous operation since then (Sztipanov et al., 2019). Detailed information on

instrument calibration can be found in Sztipanov et al. (2019). Hereinafter, this station will be referred to as Troll.

The last station used in this study is Concordia, which is located at 3233 m a.s.l. on the Dome C in the Antarctic Plateau (Fig. 1). At the Concordia station, daily TOC was obtained using the SAOZ instrument (Système d'Analyse par Observation Zénithale, Pommereau and Goutail, 1988) located at 75.1° S, 124.4° E. TOC data from this instrument have been available since January 2007. This instrument is included in the International Network for the Detection of Atmospheric Composition

Change (NDACC; De Mazière et al., 2018) and is also part of the French research infrastructure dedicated to aerosols, clouds, and trace gases (ACTRIS). The SAOZ is a passive remote-sensing device designed to detect sunlight scattered from the zenith sky. Ozone is observed within the visible Chappuis band range (450–550 nm), where its absorption cross-sections show only a weak dependence on temperature. This allows for accurate monitoring of stratospheric constituents during twilight, both at sunrise and sunset, within a solar zenith angle (SZA) range of 86° to 91° (Hendrick et al., 2011; Pazmiño et al., 2023).

**2.2 Satellite TOC data**

To supplement the ground-based observations, the TOC product OMTO3 V003 derived from the Ozone Monitoring Instrument (OMI) overpass data was used (https://avdc.gsfc.nasa.gov/pub/data/satellite/Aura/OMI/V03/L2OVP/OMTO3/). This TOC product is retrieved from the enhanced TOMS version-8 algorithm developed by NASA (Bhartia and Wellemeyer, 2002). As seen from Fig. 2a–c, the ground-based TOC observations at remote Antarctic stations have numerous gaps in the record

throughout the study period. Therefore, the missing daily TOC data were supplemented using the OMI satellite data. If a given month still had more than five missing values after the OMI data were added, it was not included in the analyses. Unlike the Troll and Concordia stations, which have been operating since 2007, the Marambio station only provided ground-based




measurements between February 2010 and January 2020. Therefore, in order to maintain a consistent study period (2007–
2023), TOC data between January 2007 and January 2010, and February 2020 and December 2023 were supplemented from
OMI. A study by Čížková et al. (2019), which compared TOC from the B199 instrument and satellite data from OMI (OMTO3
product) in the period 2011–2013, concluded that among the available satellite data products, the OMTO3 product was in the
best agreement with the B199 Brewer spectrophotometer measurements. This data product generally has a good agreement
with a mean difference of less than 1 % (Čížková et al., 2019). For the SAOZ instrument, the mean difference from satellite
products in polar regions ranges between +1 and +2 % (Hendrick et al., 2011).

## 2.3 MERRA-2 TOC data

In this study, reanalysis data from the second version of the Modern-Era Retrospective analysis for Research and Applications
(MERRA-2), with a horizontal grid resolution of 0.625° × 0.5°, have been utilised for the addition of annual cycles and spatial
analysis (https://gmao.gsfc.nasa.gov/reanalysis/merra-2/). MERRA-2 is an atmospheric reanalysis produced by the Global
Modelling and Assimilation Office of NASA (Gelaro et al., 2017). Studies utilising ozone data from MERRA-2 have shown
that these datasets exhibit high quality when evaluated against both satellite and ground-based observations (e.g., Rienecker et
al., 2011; Wargan et al., 2017; Zhao et al., 2017, 2019, 2021; Fioletov et al., 2023). According to Zhao et al. (2021), the bias
between MERRA-2 and Brewer world reference instruments ranges from −0.27 % to 1.05 %, based on hourly data from 1999
to 2019, with the standard deviation of monthly differences remaining below 1.2 %.
Starting in October 2004, MERRA-2 began integrating ozone profile data from the Microwave Limb Sounder (MLS) along
with total column measurements provided by the Ozone Monitoring Instrument (OMI) (Wargan et al., 2017). Both instruments
are components of NASA's Earth Observing System Aura satellite, which was launched in 2004. SBUV and OMI retrieve
ozone concentrations by detecting backscattered solar radiation in the atmosphere, whereas MLS derives its observations from
thermal microwave emissions (Wargan et al., 2017). MERRA-2 provides a continuous and homogeneous ozone record
spanning from 1980 to 2023, with a temporal resolution of one hour. The study by Wargan et al. (2017) offers strong support
for the use of MERRA-2 data in scientific research focused on stratospheric and upper tropospheric ozone.

## 3 Multiple linear regression model

To assess the trend in Antarctic TOC and the factors affecting it, the multiple linear regression model developed as part of the
Long-term Ozone Trends and Uncertainties in the Stratosphere (LOTUS) activity was applied. This LOTUS regression was
tested with several ozone datasets, described in detail in SPARC/IO3C/GAW (2019). The method is suitable for both point
station TOC data (Bernet et al., 2023) and spatial data. This study uses the model version 0.8.0 (USask ARG and LOTUS
Group, 2017; Bernet et al., 2023), which was extended with additional predictors to increase the description of the ozone
variability (see Section 4.1). The regression function has the following form:

$$y(t) = a + b \cdot t + \sum_{n=1}^{4} \left( c_n \cdot \sin\left(\frac{2\pi}{l_n} \cdot t\right) + d_n \cdot \cos\left(\frac{2\pi}{l_n} \cdot t\right) \right) + \sum_{n=1}^{m} (\beta_n \cdot X_n), \quad (1)$$





where $y(t)$ is the modelled TOC time series, t is the time vector of monthly means, constant intercept $a$ and linear term $b$. The
seasonal cycle is accounted for by adding annual oscillations and some overtones ($l_n$= 12, 4 and 3 months) with fitted
coefficients $c_n$ and $d_n$. Because of the incomplete seasonal cycles resulting from missing winter data, the 6-month period was
not used in the model as it did not provide additional explained variability. Furthermore, $m$ explanatory variables $X_n$ and their
fitted coefficients $\beta_n$ were included in the regression to explain the natural variability of ozone.

The regression model was applied to monthly averages from the compiled daily TOC time series at three Antarctic stations
(see section 2.2). Months with more than 5 missing measurements were not included. A similar approach was also applied in
the study by Bernet et al. (2023), where months with fewer than 25 measurement days were excluded. This ensures a
representative TOC average for the given month. This means that for all three stations, monthly averages are excluded for
April to August. The trend analysis starts in 2007, as data from Troll and Concordia are available from January 2007.

Furthermore, the LOTUS regression was applied to each grid (0.625° × 0.5°) in the 40–90° S region using TOC from MERRA-
2 data. These data are complete for all months, so oscillations with $l_n$ = 12, 6, 4, and 3 were included (in section 4.6).

## 3.1 Predictor choice

The primary objective of the LOTUS regression was to estimate stratospheric TOC trend profiles using an extensive range of
global satellite datasets (SPARC/IO3C/GAW, 2019). The default predictors initially included the El Niño–Southern
Oscillation (ENSO) (e.g., Oman et al., 2013), the Quasi-Biennial Oscillation (QBO) (e.g., Baldwin et al., 2001), the 10.7 cm
solar flux, and aerosol optical depth (AOD) at 3.0 cm wavelength (e.g., Solomon et al., 1998). AOD was not included in this
study because its influence on TOC is relevant mainly for important volcanic eruptions (e.g., Solomon et al., 1998).
Furthermore, AOD did not contribute any additional explained variability to the model. It was also not included in the final
version of the model in a study by Bernet et al. (2023). LOTUS regression can also be used to assess TOC at individual stations,
where additional explanatory predictors are included (Van Malderen et al., 2021; Bernet et al., 2023).



**Table 1: Predictors for use in the multiple linear regression model. The default LOTUS predictors are marked with an asterisk (\*).**

| Predictor | Full predictor name | Data and source |
|---|---|---|
| EHF | Mean eddy heat flux | Heat flux at 100 hPa from MERRA-2 reanalysis, averaged over 45 to 75 N (deseasonalized). https://acd-ext.gsfc.nasa.gov/Data_services/met/ann_data.html |
| ENSO* | El Niño–Southern Oscillation | Multivariate ENSO index (version 2) derived from five surface variables. https://psl.noaa.gov/enso/mei/data/meiv2.data |
| IOD | Indian Ocean Dipole | Indian Ocean Dipole is represented by the Dipole Mode Index (DMI). https://psl.noaa.gov/gcos_wgsp/Timeseries/Data/dmi.had.long.data |
| Solar* | Solar flux | Adjusted solar index at 10.7cm from OMNI. https://omniweb.gsfc.nasa.gov/form/dx1.html |
| T100 | Stratospheric temperature | Deseasonalized temperature at 100 hPa from MERRA-2 reanalysis at each station (deseasonalized). https://disc.gsfc.nasa.gov/ |
| QBO* | Quasi-Biennial Oscillation | Four principal components of equatorial wind at 7 pressure levels (70, 50, 40, 30, 20, 15, 10hPa). https://acd-ext.gsfc.nasa.gov/Data_services/met/qbo/QBO_Singapore_Uvals_GSFC.txt |

In addition to the default LOTUS predictors, several other relevant predictors are used in this study, all of which are described in Tab. 1. The 100 hPa temperature (T100) was used as the first additional predictor. This predictor represents the temperature

in the lower stratosphere and plays a significant role in the dynamics of TOC variability (Ningombam et al., 2020). Unlike the study by Bernet et al. (2023), the tropopause pressure (TropP) predictor was not used due to its strong correlation with T100 ($r = 0.93$–$0.98$).

Inclusion of the Brewer–Dobson Circulation (BDC) was also necessary, as it plays a key role in explaining the natural variability of ozone, particularly at high latitudes (Plumb, 2002; Bernet et al., 2023). The intensity of the BDC can be assessed

through the upward propagation of planetary waves, quantified by the meridional eddy heat flux (EHF) (Gabriel and Schmitz, 2003). Consequently, the mean EHF at the 100 hPa pressure level, averaged over 45 to 75° S, was used as an indicator of BDC strength. Circulation patterns such as the IOD, which has received little attention regarding its influence on Antarctic TOC, have also been used.

Adjustments were made to the QBO, which is the initial LOTUS regression predictor. According to the methodology used in

the study by Bernet et al. (2023), 7 pressure levels between 70–10 hPa were used, from which the principal component analysis was calculated. The first four principal components were then used in the study (will be designated as QBOa – QBOd). The Quasi-Biennial Oscillation (QBO) arises from oscillations of equatorial stratospheric winds, which influence ozone concentrations from the tropics to the polar regions (Wang et al., 2022). At higher latitudes, however, the amplitude, phase, and frequency of these oscillations can vary (Damadeo et al., 2014). Therefore, it is preferable to use the principal components



of QBO at seven pressure levels in higher latitudes rather than relying on the direct QBO time series (Damadeo et al., 2014; SPARC/IO3C/GAW, 2019; Bernet et al., 2023). No statistically significant trend ($p < 0.05$) was detected in the time series of predictors (Fig. S1).

Multicollinearity between predictors was assessed using variance inflation factors (VIF), as shown in Tab. 2. For all predictors, the VIF value ranges between 1–2. Multicollinearity is not considered problematic when the VIF is below 5, as suggested by

Schuenemeyer and Drew (2010). Concrete Pearson correlation coefficients for individual stations are shown in Fig. S2.

**Table 2: Variance inflation factors (VIF) for selected predictors included in the regression model**

| Predictors | Marambio | Troll | Concordia |
|------------|----------|-------|-----------|
| EHF | 1.11 | 1.06 | 1.18 |
| ENSO | 1.30 | 1.33 | 1.30 |
| IOD | 1.60 | 1.69 | 1.61 |
| SOLAR | 1.17 | 1.19 | 1.19 |
| T100 | 1.09 | 1.05 | 1.21 |
| QBOa | 1.26 | 1.27 | 1.27 |
| QBOb | 1.05 | 1.09 | 1.06 |
| QBOc | 1.60 | 1.59 | 1.62 |
| QBOd | 1.41 | 1.44 | 1.40 |

## 4 Results and discussion

### 4.1 Time series comparison and validation with OMI and MERRA-2

Figure 2 compares ground-based TOC measurements, MERRA-2 reanalysed data, and satellite overpasses at three Antarctic stations. The ground-based TOC measurements, shown in the first row, are supplemented with the OMI and MERRA-2 data. TOC variation at individual stations well demonstrates the occurrence of a seasonal cycle, which is most pronounced in the case of the Troll station. At the Marambio station, the effect of ozone depletion is not as great as at the Troll and Concordia stations. This is due to Marambio's location at the edge of the polar vortex and the subsequent frequent alternation of polar and

subpolar air masses (Čížková et al. 2023). The Troll station achieves the lowest winter TOC averages (about 150–190 DU) annually.

The second row (Fig. 2d–f) presents a comparison of monthly TOC means between ground-based (GB) measurements, satellite overpasses and MERRA-2 reanalysed data. All stations' average relative deviations between GB and OMI data are within 3 %. The lowest deviation is at the Troll station (-0.28 %), followed by Marambio (-1.30 %) and the Concordia station (-2.42

%). These results align with the satellite measurement uncertainties of around 2 %, as reported by Bodeker and Kremser (2021). Similar deviations between GB and OMI data have been observed in the Arctic (Bernet et al., 2023). Larger differences can be seen for individual months, but they are never greater than ±9 %. The differences between GB and MERRA-2 data are very




similar; the mean bias is highest at the Concordia station (-4.31 %), and it is within ±1 % at the other stations. The monthly

differences are highest at the Concordia station, but do not exceed ±10 %, which is comparable to the OMI data.

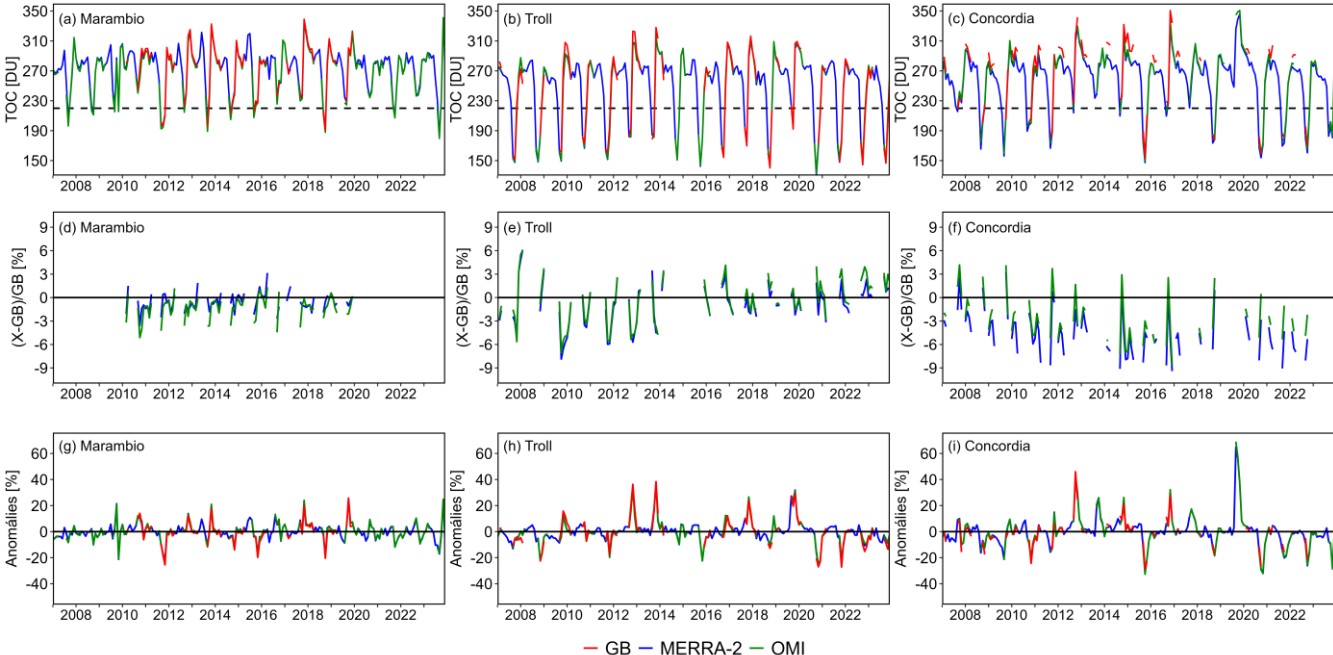


**Figure 2: Comparison of ground-based total ozone column (TOC) data with satellite overpasses from OMI and MERRA-2 data at the Marambio, Troll and Concordia stations for the period 2007–2023. The first row (a–c) shows monthly means of the ground-based data (red) together with OMI (green), and MERRA-2 (blue) monthly means. The second row (d–f) shows monthly relative differences between ground-based data and other datasets (blue with MERRA-2 and green with OMI). The third row (g–i) shows**
**relative anomalies for each dataset (red - the ground-based data, blue - MERRA-2, and green - OMI), which are defined as the deviation of each month from the monthly mean climatology (2007–2023) of the respective dataset. The dashed lines in panels (a–c) show TOC=220 DU.**

The monthly TOC anomalies represent deviations from the 2007 – 2023 monthly means (Fig. 2g–i). This implies that at the

Marambio station, TOC naturally varies in the range of ~20 %, at the Troll and Concordia stations in the range of ~40 %.
Individual data sets move within their natural variability, but some years show more significant anomalies. For example, in

the winter of 2019, Antarctica experienced a sudden stratospheric warming, causing generally higher TOC (e.g., Safieddine et

al. 2020), which resulted in the strong positive anomaly (~60 %) at the Concordia station in September 2019.

Section 2.2 clarifies that the time series of GB measurements were supplemented with satellite overpass data from OMI. Fig.

S3 shows monthly medians and quantiles of TOC at the three stations. At Marambio, the lowest median TOC was recorded in
September (207.52 DU) and the highest in December (308.31 DU), with the greatest variability observed from September to

November. Troll showed the lowest median in October (156.23 DU) and the highest in December (292.40 DU), with the

highest variability in November and December, and the lowest in September–October and January–April. Concordia exhibited

a similar seasonal pattern, with the lowest TOC in October (192.93 DU) and the highest in December (296.42 DU), and





increased variability during the austral spring. Outliers in September and October at Troll and Concordia likely reflect a weaker

polar vortex and elevated TOC levels in 2019.

## 4.2 Marambio

Figure 3 shows the results of the linear regression model for the Marambio station, where the adjusted coefficient of determination ($R^2_{adj}$) explains 94 % of the TOC variance. The residuals mostly lie within the 5 % range, with the Shapiro-Wilk test (Shapiro and Wilk, 1965) confirming the normal distribution of the residuals and the Durbin-Watson test (Durbin and

Watson, 1950) indicating no autocorrelation of the residuals (the value of the test statistic was close to 2). A normal distribution and no autocorrelation of the residuals were also demonstrated for the Troll and Concordia stations. At the Marambio station, an increasing statistically significant trend of 3.43 ±3.22 DU/decade was detected.

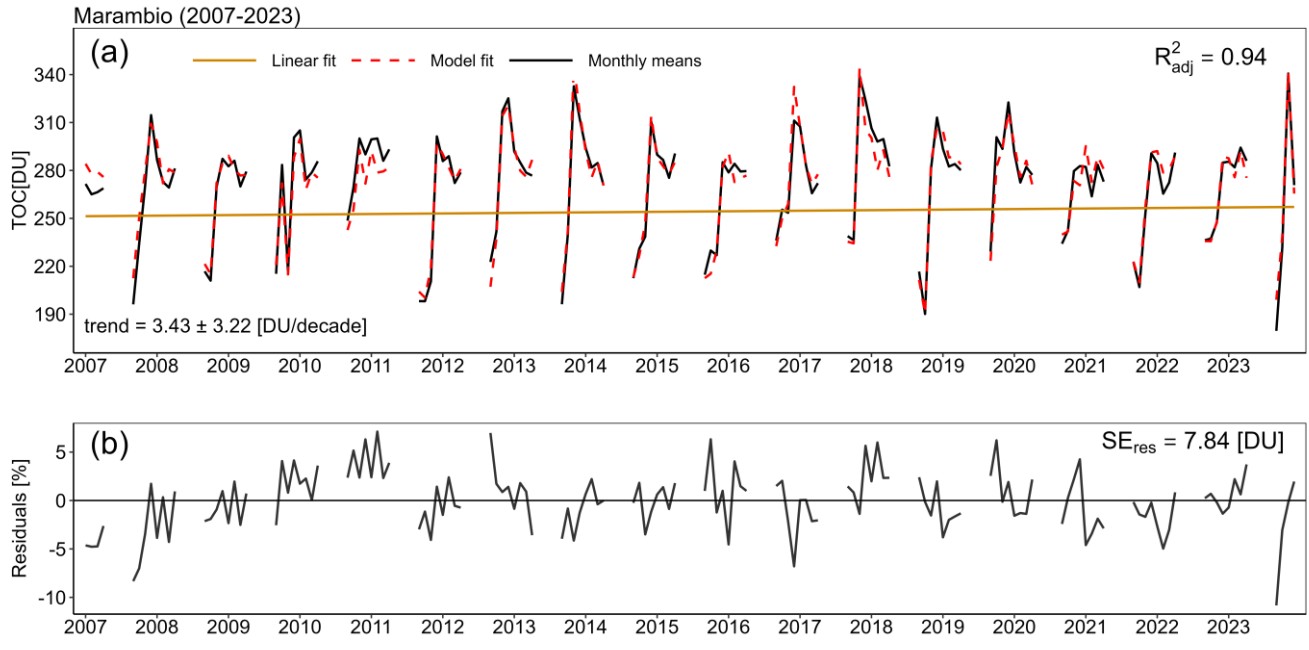

**Figure 3: Regression fit (a) and residuals (b) at Marambio.**

Figure 4 shows that a large part of the TOC variability at the Marambio station can be explained by T100. This predictor causes the change in TOC between -50 and 60 DU. EHF affects TOC variability between -4 and 2 DU, and it is clear that in 2019, when the SSW occurred at Marambio, EHF caused a TOC decrease of approximately 4 DU. The TOC variability explained by ENSO shows that La-Niña events, for example, in 2011 (Bastos et al., 2013), increase the TOC amount, while El-Niño events, such as in 2015 (Santoso et al., 2017), decrease TOC at Marambio. More detailed information on the influence of

ENSO on Antarctic TOC has been provided in a study by Lin and Qian (2019). The IOD oscillation has only a negligible (±2 DU) effect on TOC at Marambio. In the case of the solar factor, 11-year solar cycles are clearly visible, with TOC decreasing





by almost 3 DU during solar maxima and TOC increasing slightly (1 DU) during solar minima. The last of the QBO predictors affects TOC variability by up to ±5 DU.

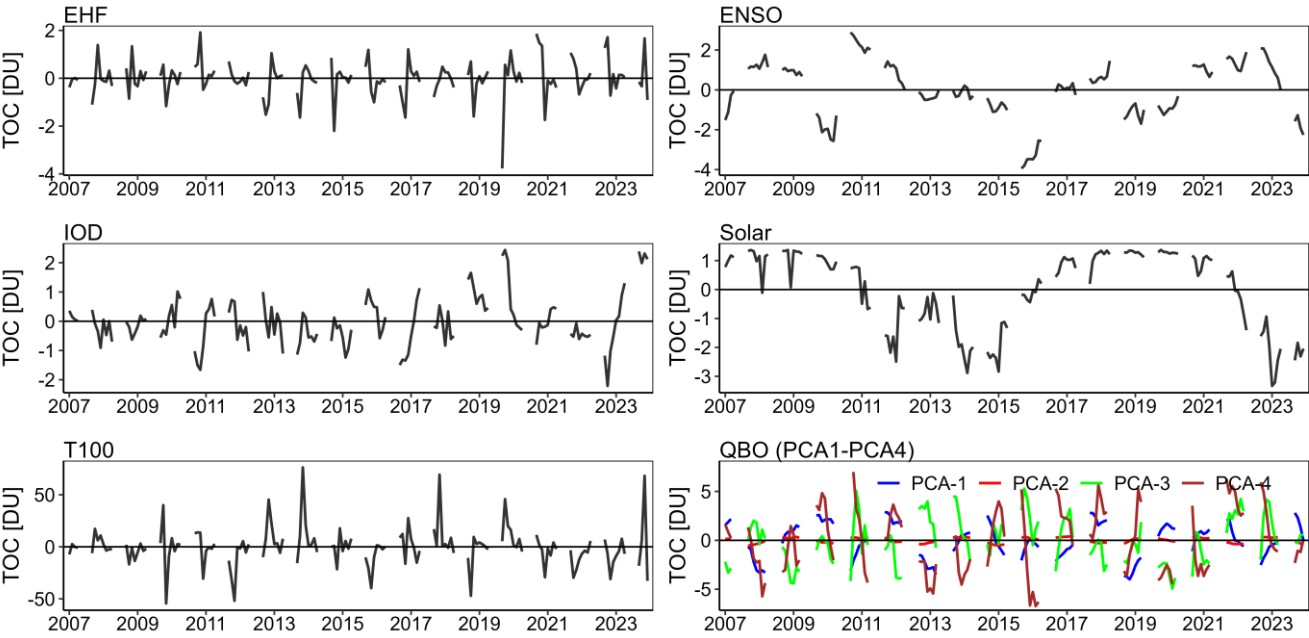

Figure 4: Predictor contribution ($\beta_n \cdot X_n$, with coefficient $\beta_n$ and predictor $X_n$) at Marambio.

The coefficients of the individual predictors and their statistical significance for the annual regression fit at Marambio are shown in Fig. 5a. For a direct comparison, the coefficients have been standardised. Brunner et al. (2006) define the standardised coefficients $\beta_{std}$ as the percentage change in TOC associated with a 1σ change in each predictor:

$$\beta_{std} = \beta \cdot \frac{\sigma_X}{\bar{y}} \cdot 100 \tag{2}$$

where $\beta$ denotes the predictor coefficient, $\sigma_X$ the standard deviation of the predictor $X$, and $\bar{y}$ the mean TOC. The predictor T100 exhibited the largest statistically significant ($p<0.05$) contribution to the variability of TOC (Fig. 5a). Other statistically significant predictors include only QBOd. Other predictors have a statistically insignificant effect on the TOC variability at Marambio.





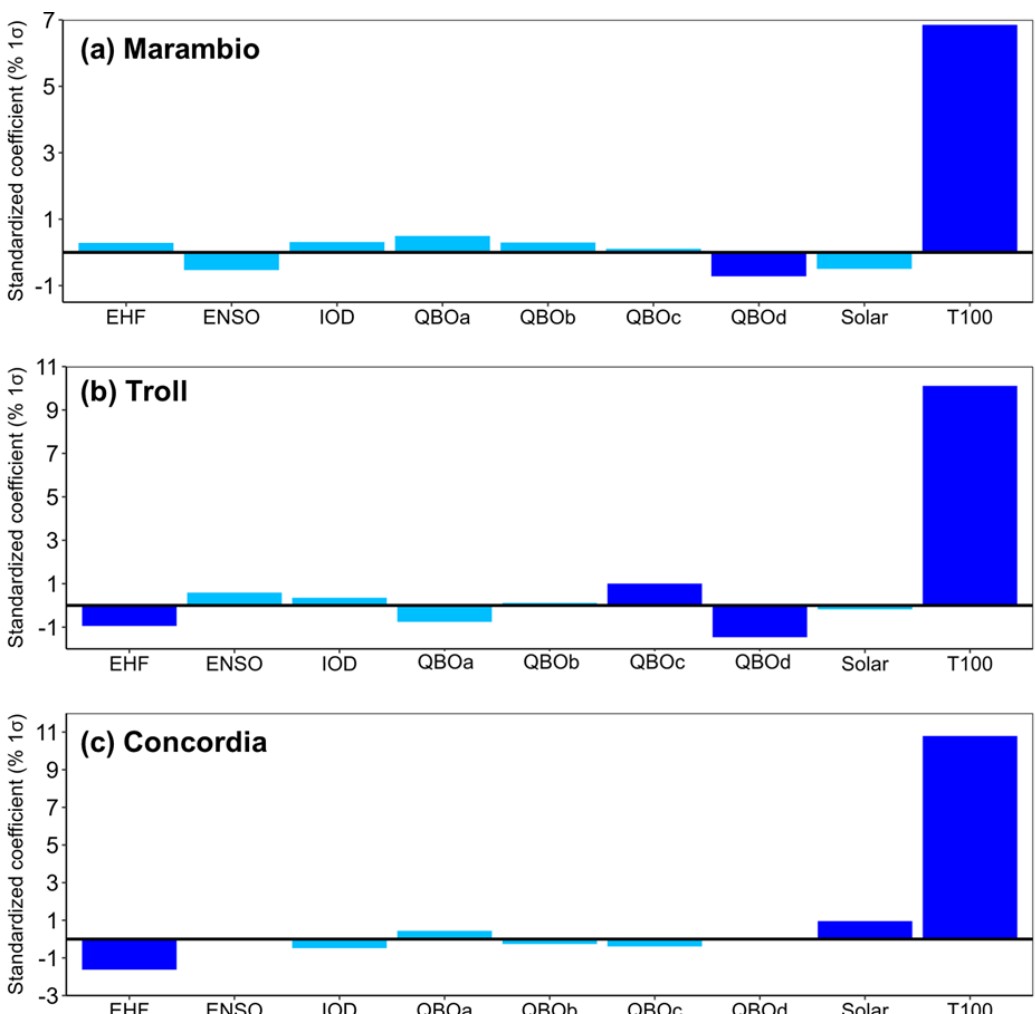

**Figure 5: Predictor contributions to the annual regression fit at Marambio (a), Troll (b) and Concordia (c). Standardised coefficients indicate the percentage change in TOC associated with a one standard deviation change in the predictor. Light blue bars denote predictors whose effect on ozone is not statistically significant (p-value of the coefficient <0.05).**

### 4.3 Troll

The results of the linear regression model for the Troll station, where the adjusted coefficient of determination ($R^2_{adj}$) explained 97 % of the TOC variance, are shown in Fig. 6. The residuals are mostly within ±10 %. In the case of the Troll station, a decreasing but statistically insignificant trend of -1.09 ±3.91 DU/decade was found. The larger percentage of explained variability at the Troll station compared to Marambio may be due to the lower interannual variability of the monthly mean TOC.



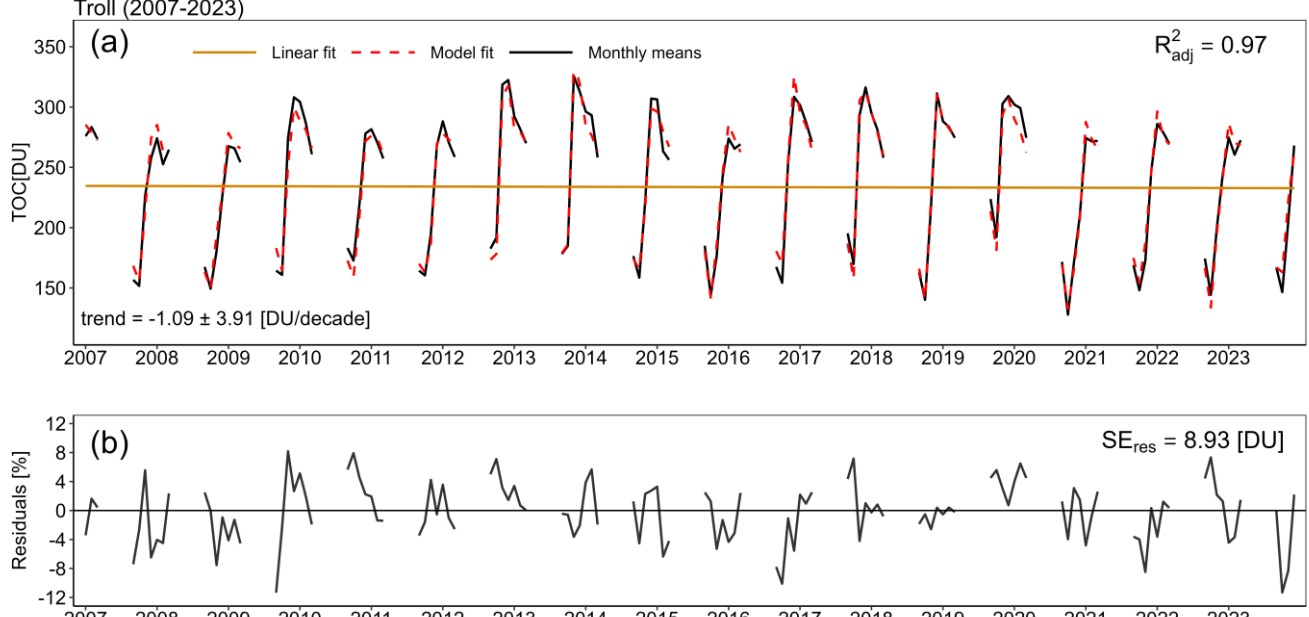

**Figure 6: Regression fit (a) and residuals (b) at Troll.**

At the Troll station, TOC variability is mainly influenced by T100, as is the case at the Marambio station (Fig. S4). Temperature in the lower stratosphere causes a change in TOC between -50 and 100 DU. The largest TOC loss due to temperature in the lower stratosphere occurred in 2020, when the Antarctic stratosphere was very cold during the winter and spring months. On the contrary, in October 2012 and 2013, ozone increased by ~80 DU at the Troll station due to the disruption of the polar

vortex by heat fluxes (Klekociuk et al. 2014, 2015). EHF and QBO between -5 and 10 DU also had a more significant influence on TOC; however, the influence of other predictors was rather negligible. Interestingly, at Troll, predictors such as EHF and ENSO behave oppositely to those at the Marambio station. This leads to the possible conclusion that on the opposite coastal parts of the Weddell Sea, some predictors may influence TOC with opposite effects.

Figure 5b shows the standardised coefficients of determination for each predictor and their statistical significance. At the Troll

station, T100 has the largest and statistically significant influence on TOC variability. This influence is approximately 4 % higher than at the Marambio station. Other statistically significant predictors include $QBO_{c,d}$ and EHF. The rest of the predictors are not statistically significant in the case of the Troll station. Compared to the Marambio station, a greater influence of QBO and EHF can be observed.

## 4.4 Concordia

The last station for which a linear regression model was applied is Concordia. At this station, the adjusted coefficient of determination ($R^2_{adj}$) explained 96 % of the TOC variance (Fig. 7a). The trend (1.09 ±4 DU/decade) was not statistically significant. The residuals are mostly within ±9 %.



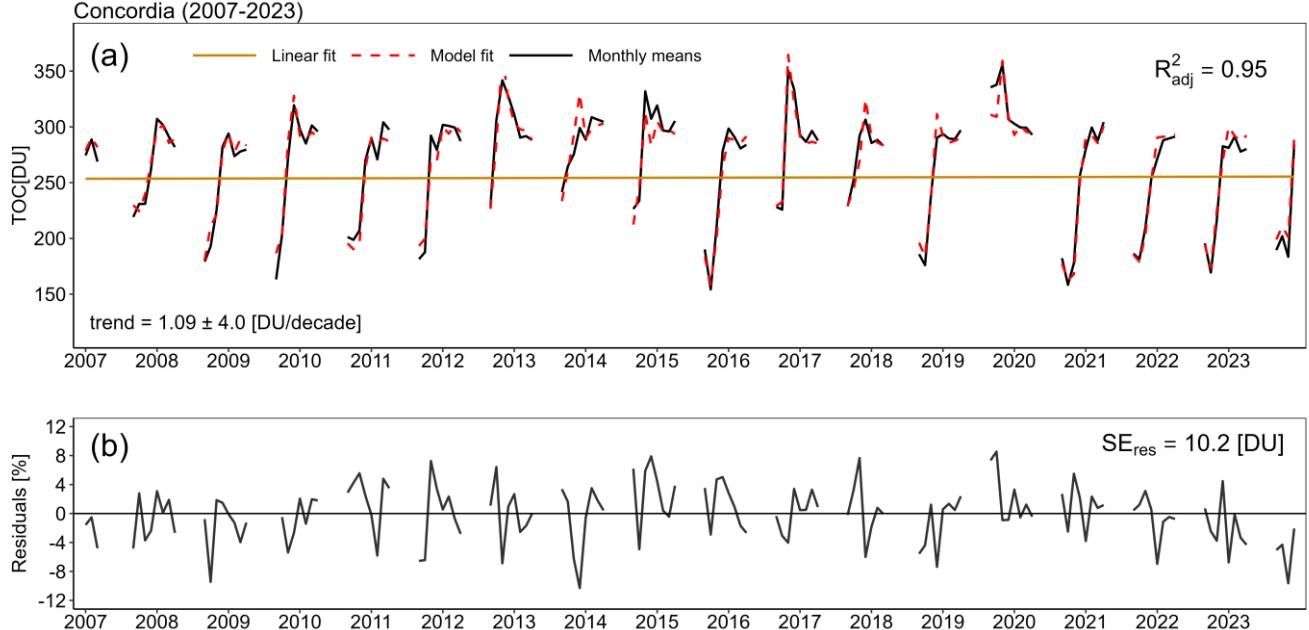

**Figure 7: Regression fit (a) and residuals (b) at Concordia.**

Figure S5 shows that, also at Concordia, the variability of TOC is driven mainly by the temperature in the lower stratosphere (T100). In 2019, TOC increased by approximately 20 DU due to EHF. This predictor behaves similarly to the Troll station, but in the opposite way compared to Marambio. The other included predictors contribute to the TOC variability by up to 5 DU. The solar factor is worth mentioning, as it exhibits the opposite behaviour here than at the Troll and Marambio stations. The solar factor also explains a greater amount of TOC variability than at the previous two stations.

Based on the standardised coefficients of determination for individual predictors, the T100 predictor has the greatest and statistically significant influence on TOC variability at the Concordia station (Fig. 5c). The influence of this predictor on TOC is comparable to that of the Troll station. Other statistically significant predictors are EHF and solar. The EHF predictor negatively affects TOC, which is different from the Marambio station, where this predictor has a statistically insignificant but positive effect on TOC. Also worth mentioning is the solar factor, which is statistically significant only at the Concordia
station.

A similar study was conducted for European subpolar and polar stations, where their ground-based time series for the period 2000–2020 are assessed (Bernet et al., 2023). The study by Bernet et al. (2023) used QBO, Solar, ENSO, EHF, TropP and T50 as predictors. At the Oslo, Andoya and Ny-Ålesund stations, the TropP predictor had the greatest and statistically significant influence on TOC variability, followed by T50 (stratospheric temperature at 50 hPa) at the Oslo and Andoya stations and EHF
at the Ny-Ålesund station. However, this study shows that in Antarctica, the temperature in the lower stratosphere (T100) has the greatest influence on TOC variability. TropP was not used precisely because of the high correlation with T100. Interestingly, T100 has a slightly different influence on TOC variability in individual years at each station. However, in October



2013, TOC increased uniformly at all stations due to T100. The predictor time series (Fig. S1) shows that in October 2013, T100 had one of the highest positive anomalies at all three stations. According to the study by Klekociuk et al. (2015), initially

experienced anomalously low winter temperatures in the polar stratosphere and a concomitant strong and stable polar vortex, supporting the potential for strong ozone depletion in 2013. However, from late August onwards, anomalous warming of the polar vortex occurred, limiting ozone depletion during the spring and leading to a relatively early ozone hole breakdown (Klekociuk et al. 2015).

## 4.5 Trends during SON

Trend analysis was also performed for the spring months (SEP, OCT, NOV) at three Antarctic stations. In this case, LOTUS regression was not used due to multicollinearity between predictors at the monthly scale (VIF > 5; not shown). Previous studies suggest that signs of ozone recovery can be observed over Antarctica during September since 2000 (Solomon et al., 2016; Weber et al., 2018, 2022), with the magnitude of the Antarctic ozone anomaly gradually increasing to reach its maximum in late September and early October (Weber et al., 2018). Pazmino et al. (2018) evaluated TOC trends within the southern polar

vortex and found that the largest trends and highest significance were found for September in the period 2001–2017, with a trend value of $1.84 \pm 1$ DU. TOC trends in the spring months are of great interest in the polar regions, as these regions experienced the largest ozone depletion in the period before 2000 (e.g. Solomon, 1999).

Trend analysis in Fig. 8 showed that in the period 2007–2023, the trend at all stations in all three months is statistically insignificant (2sigma is higher than the trend value at all stations). The reason for the statistically insignificant trend may be

the larger and more persistent spring ozone anomalies that were observed between 2020 and 2023 (Kessenich et al. 2023; Kozubek et al. 2024). Interestingly, the trends for October and November are statistically insignificantly decreasing at all stations, while the trend for September is statistically significantly increasing. The different behaviour of the trends may be related to the later occurrence of the ozone anomaly, which in 2015 and 2020 reached record size only in October–December (Stone et al., 2021). The persistence of the polar vortex during these years was likely enhanced by external factors: the 2015

eruption of the Calbuco volcano (Zhu et al., 2018) and extensive biomass burning in Australia in 2019–2020 (Salawitch and McBride, 2022). It is important to note that determining precise ozone trends in the southern polar stratosphere is difficult due to saturation of ozone loss, i.e. complete or near-zero ozone destruction in the lower stratospheric layers, mostly at an altitude of 13–21 km (Kuttippurath et al., 2018). At the higher latitudes (Troll and Concordia), saturation of ozone loss may obscure signs of recovery, in contrast to the vortex edge latitudes (Marambio).





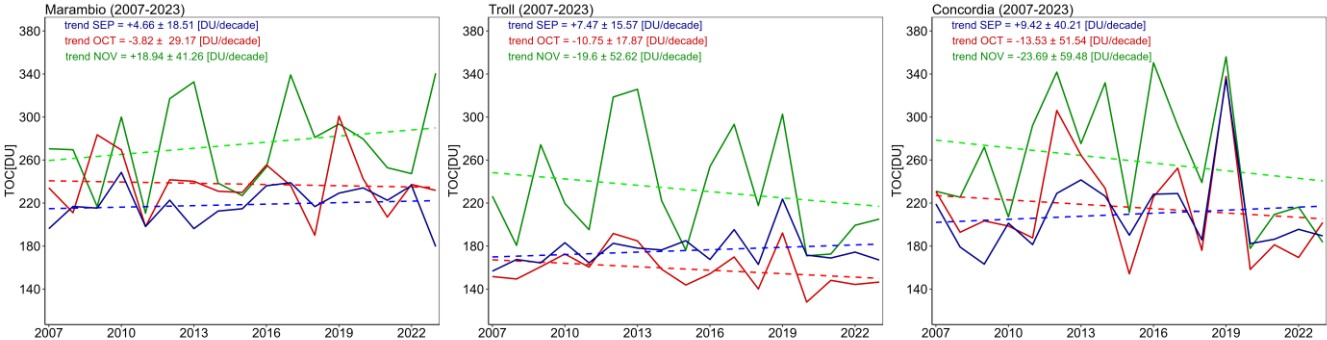

**Figure 8: TOC variability for September (blue), October (red) and November (green) supplemented with the linear trend (dashed) for Marambio (left), Troll (middle) and Concordia (right) in the period 2007–2023.**

### 4.6 Spatial effect of factors affecting TOC

The linear regression from equation 1 was calculated for each grid (0.625° x 0.5°) in the 40–90° S area in the period 2007–2023 based on TOC from the MERRA-2 reanalysis (Fig. 9). The T100 and TropP predictors had individual values for each grid point, while the other predictors (e.g., ENSO, IOD, PSC, etc.) had the same value for the entire region. All months have been included in the assessment; therefore, multiple linear regression was calculated with the periods l = 12, 6, 4, and 3.

The spatial representation of the LOTUS regression for each grid point allows for a unique representation of the model results, as the model represents the spatial distribution of the effects of individual predictors, which are expressed using standardised coefficients of determination. The spatial distribution of the adjusted coefficient of determination shows that the used LOTUS model best represents the real TOC series over the Queen Maud Land and the Weddell Sea, where $R^2_{adj}$ approaches 0.97. Furthermore, the model performs well over the Indian Ocean and Australia ($R^2_{adj} > 0.95$). It is less representative of TOC in the Atlantic Ocean sector and in the marginal parts of Antarctica, which may be caused by the increased TOC and dynamical variability of the edge of the polar vortex. However, even in the subantarctic areas, the adjusted determination coefficient explains more than 70 % of TOC variance.

The EHF predictor has a positive, statistically significant effect on TOC over the ocean west of Antarctica, but it affects TOC negatively over East Antarctica. The influence of ENSO on the TOC over Antarctica is statistically insignificant, except in the sector near Australia, which has a statistically significant positive influence. A study (Lin and Qian, 2019) shows that the strongest ozone anomalies over Antarctica occur one year after El Niño and La Niña, which have a dipole structure between the upper and lower stratosphere. The IOD has a statistically insignificant effect on TOC in Antarctica, while it has a positive effect in the surrounding oceans. The solar factor has a statistically significant positive effect on TOC over East Antarctica and a negative effect in the Atlantic Ocean sector. The strongest and statistically significant positive effect on TOC over Antarctica is characterised by the temperature in the lower stratosphere (T100) as seen on individual stations considered in this study (sections 4.2, 4.3, and 4.4).





The last predictor evaluated was the QBO, represented by the first four principal components (QBOa – QBOd) calculated from seven pressure levels between 10 and 70 hPa. The QBOa has a statistically significant positive effect on TOC over central and West Antarctica, whereas the QBOd has a significant negative effect over West Antarctica. QBOb affects TOC with a statistically significant negative in Antarctica over a small area near Concordia station and over the Southern Ocean. The Antarctic TOC was not significantly affected by QBOc, as there is only a small area over the Amundsen Sea where a significant

negative effect was found.

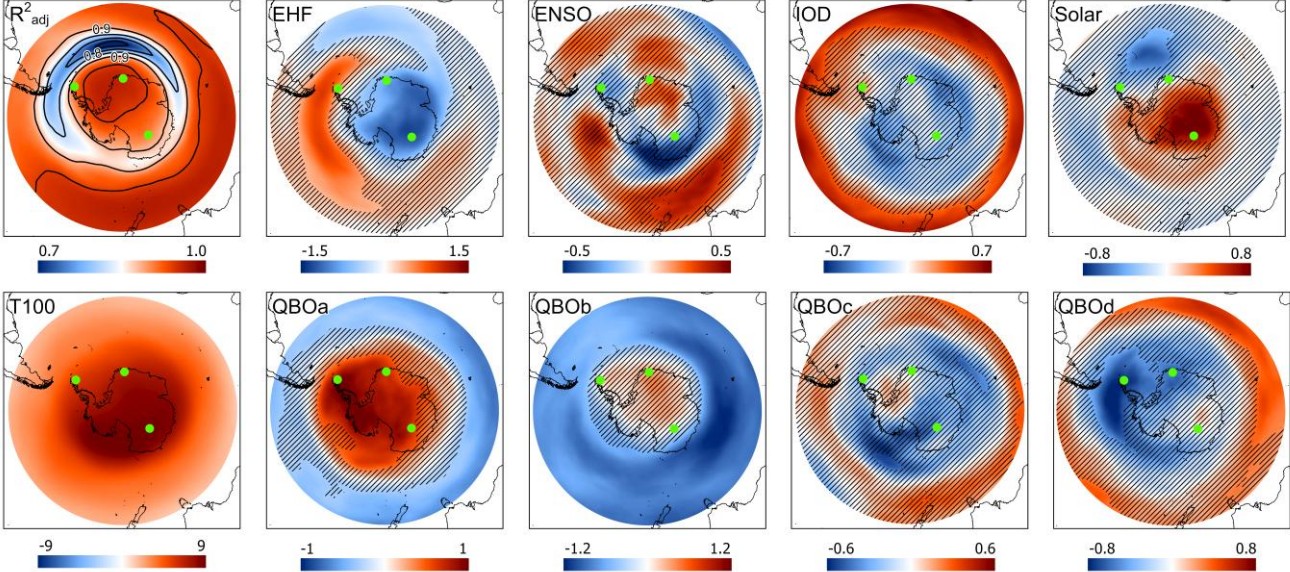

**Figure 9: Spatial distribution of the adjusted coefficient of determination ($R^2_{adj}$) and the effect of individual predictors shown using standardised coefficients of determination. The unshaded area is statistically significant (p<0.05).**

### 4.6.1 Comparison of September 2019 and 2020

The spring seasons of 2019 and 2020 were very different in Antarctica in terms of stratospheric dynamics and TOC evolution. In the first half of September 2019, a minor sudden stratospheric warming (SSW) occurred in the Southern Hemisphere (Liu et al., 2022). The SSW was caused by a quasi-stationary planetary wave of zonal wavenumber 1 (Liu et al., 2022). In 2019, an atypical warming of the polar vortex lessened ozone depletion in early September, resulting in the second smallest ozone hole ever recorded (Jonson et al., 2023). Studies by Wargan et al. (2020) and Safieddine et al. (2020) also concluded that a reduced

ozone hole area was indeed observed in Antarctica in 2019. In contrast, during the spring of 2020, the polar vortex was unusually strong and persistent. A study by Lin et al. (2024) reported that it was the strongest and coldest event since 1979 in the middle to lower stratosphere on average from October to December, resulting in a significantly lower TOC during this period.





Figure 10 compares these two contrasting months in terms of the impact of individual LOTUS regression predictors. Similarly
to the previous results presented in this study, lower stratospheric temperature (T100) had the greatest impact on the TOC. In
September 2019, the relatively high T100 led to a more than 100 DU increase in TOC in East Antarctica. These results are
closely related to the observed temperature contributions to TOC variability in September 2019 and 2020 at individual stations.
For example, in September 2019, there was a 60 % increase in TOC compared to the long-term average observed at the
Concordia station. The area of significant TOC increase due to higher stratospheric temperatures in September 2019
corresponds to the stratospheric disturbance by a planetary wave with wavenumber 1 (Mitra et al., 2021; Liu et al., 2022).
Based on MERRA-2 reanalyses, on 5–11 September 2019, a strong planetary wave during the SSW caused a remarkable
temperature increase of 50.8 K (Yamazaki et al., 2020; Lim et al., 2021). This exceeded even the 2002 Antarctic SSW event,
when the southern stratospheric polar vortex split in two and when the maximum warming was 38.5 K per week (Leroux and
Noel, 2024). In late August 2019, the rapid warming of the stratosphere led to a significant decrease in PSC densities between
50 and 150 hPa (Leroux and Noel, 2024). The unusually high temperature in the polar Antarctic stratosphere leads to a
slowdown in the heterogeneous reactions producing amounts of activate Cl that will destroy ozone in PSC, which suppresses
the formation of the Antarctic ozone anomaly during the Antarctic spring (Liu et al., 2022).

The contribution of the EHF is also significant, leading to an increase in TOC, especially in East Antarctica, by up to 15 DU
in September 2019. In the period from August to September 2019, the EHF was the strongest since 1979 (Shen et al., 2020).
Based on further decomposition of the EHF, it was found that the EHF is dominated by a planetary wave with wave number
1. This wave persisted for approximately 1 month, and during its maximum, it exceeded 10σ with respect to the period 1979–
2019 (Shen et al., 2020). The stronger EHF (which is used as a proxy for BDC here) led to a stronger ozone transport from
lower latitudes to polar regions (Rao et al., 2003, 2004). Planetary Rossby waves, which drive the BDC and can cause SSWs,
represent the largest changes in stratospheric circulation during the winter season and significantly influence the interannual
variability of stratospheric transport (Schoeberl, 1978; Butler et al., 2015; de la Cámara et al., 2018; Baldwin, 2021). In
September 2020, a strong polar vortex blocked the transport of ozone from lower latitudes to the Antarctic stratosphere,
resulting in ~10 DU less TOC over Antarctica due to the EHF. More TOC was present at the edges of the polar vortex, which
is clearly visible near the Marambio station.

Other predictors contributed to ozone variability in these two months only by up to 5 DU. It is evident that ENSO and IOD
have opposite effects on TOC in both years, while the Solar factor is the same in both years. This implies that the low solar
activity that occurred in these years (e.g. Ishkov, 2024) leads to a decrease in TOC by approximately 2 DU over the Antarctic
Continent and, conversely, increases TOC by approximately 2 DU in the South Atlantic. In the case of the QBO predictor, the
individual components had different effects on TOC in September 2019 and 2020. QBOa led to a slight decrease in TOC over
Antarctica and an increase in TOC in the surrounding oceans under SSW conditions in September 2019; however, under the
conditions of a colder stratosphere in September 2020, this effect was the opposite. The opposite effect was also evident in the
case of QBOd, which led to a decrease in TOC in September 2019 and to an increase in September 2020, especially between





Antarctica and South America. The remaining QBOb and QBOc are characterised by the same effect on TOC regardless of different stratospheric conditions.

**Figure 10: Predictor contribution ($\beta_n \cdot X_n$, with coefficient $\beta_n$ and monthly means of predictor $X_n$) of TOC [DU] during September 2019 and 2020.**





## 5 Conclusion

The objective of this study was to assess trends in total ozone column (TOC) at Antarctic stations, including Marambio, Troll, and Concordia. Ground-based TOC time series were compared with satellite overpass observations and the MERRA-2
reanalysis. The satellite and reanalysis data agree on average within 1 % to 3 % with the ground-based time series. The ground-based measurements were supplemented then with OMI satellite data.

The LOTUS regression model was used to derive trends in the total ozone column from 2007 to 2023, the first use of this model for ground-based total ozone column data in Antarctica. In addition to the basic LOTUS predictors, additional regression predictors were examined. Lower stratospheric temperature was found to be a dominant predictor. Although T100 was the
primary driver of TOC variability at all stations, station-specific influences were also identified, with QBO components playing a significant role at Marambio and Troll, and EHF contributing significantly at Troll and Concordia. The model incorporating the selected predictors accounts for a large portion of the variability in total ozone column at the Antarctic stations, as indicated by high adjusted coefficients of determination ($R^2_{adj} > 0.94$). A statistically significant trend of 3.43 DU per decade was found at Marambio station, but no significant trends were found at Troll and Concordia stations. Spring trends for September, October
and November were statistically insignificant.

Using MERRA-2 reanalyses, the LOTUS regression model was then applied to each grid point in the 40–90° S region. The model was found to perform very well over the Southern Hemisphere, with the highest coefficients of determination ($R^2_{adj} > 0.95$) being achieved over West Antarctica and the surrounding oceans. A case study assessing the effects of individual predictors in September 2019 and 2020 found that the exceptionally warm lower stratosphere in September 2019 increased the
total ozone column by more than 100 DU, especially over East Antarctica.

## Code and data availability

The total ozone column data from Marambio are the property of the Czech Hydrometeorological Institute, Hradec Králové, Czech Republic, and are the subject of the data policy of the above-mentioned institution. Any person interested in the underlying data should contact Martin Stránik, the head of the Solar and Ozone Observatory of the Czech Hydrometeorological
Institute, Hradec Králové (martin.stranik@chmi.cz). The data from the Norwegian Troll station are available from NILU through Tove Svendby (tms@nilu.no). SAOZ data are accessible via the NDACC database (https://www-air.larc.nasa.gov/missions/ndacc/; NDACC UVVIS Working Group, 2025) as well as through the SAOZ website (http://saoz.obs.uvsq.fr/; French SAOZ Group (CNRS-UVSQ), 2025).

OMI and overpass data are available at the Aura validation centre, for OMI (OMTO3) through
https://avdc.gsfc.nasa.gov/pub/data/satellite/Aura/OMI/V03/L2OVP/OMTO3/ (NASA Goddard Space Flight Center, 2025 (last access: 03 March 2025; NOAA PSL, 2025). The NCEP Reanalysis Derived data used for tropopause predictors were provided by the NOAA/OAR/ESRL PSL, Boulder, Colorado, USA, from their website at https://psl.noaa.gov/data/gridded/data.ncep.reanalysis.derived.tropopause.html (last access: 03 March 2025; NOAA PSL,



2025). MERRA-2 data are available from NASA's Global Modelling and Assimilation Office
(https://gmao.gsfc.nasa.gov/reanalysis/MERRA-2/, EarthData, 2025). The other sources for the predictors used in the trend
model are given in Tab. 1. The R version of the data processing is available upon request by David Tichopád
(david.tichopad@mail.muni.cz).

**Author contributions**

DT, KL, and TS designed the study; LM, TS and AP provided the resources; DT and TS designed the multiple linear regression
model; DT performed the data analyses and prepared the original paper draft; and KČ, KL, TS, AP, BP and LM reviewed and
edited the paper.

**Competing interests**

The contact author has declared that none of the authors has any competing interests.

**Acknowledgement**

The authors gratefully acknowledge the Czech Hydrometeorological Institute (CHMI), Hradec Králové, Czech Republic, for
providing the total ozone column data from the Marambio station. Special thanks go to Martin Stránik, head of the Solar and
Ozone Observatory at CHMI, for facilitating access to the data. The authors are also grateful to the LAMBI laboratory and
Marambio Base staff for operating the B199 Brewer spectrophotometer, namely to Michal Janouch, Ladislav Sieger, Michael
Brohart, Martin Stráník, Peter Hrabčák, Vladimir Savastiouk, and Hector Ochoa. The authors gratefully acknowledge the
support of the Institut National des Sciences de l'Univers (INSU) of the Centre National de la Recherche Scientifique (CNRS),
the IPEV, and the Centre National d'Études Spatiales (CNES), which made the observations with the SAOZ instruments within
the French ACTRIS Research Infrastructure possible. The authors also thank the technical teams responsible for operating the
SAOZ instruments. The authors also thank the Norwegian Ministry of Climate and Environment for funding the NILU-UV
measurements at Troll/Trollhaugen.
We also acknowledge the NASA Goddard Space Flight Center for providing the OMI and overpass data (OMTO3) through
the Aura Validation Data Center. The NCEP Reanalysis Derived data used for the tropopause-related predictors were kindly
provided by the NOAA/OAR/ESRL Physical Sciences Laboratory, Boulder, Colorado, USA. We further thank NASA's Global
Modeling and Assimilation Office for making the MERRA-2 reanalysis data available.



**Financial support**

This research was funded by the projects Czech Antarctic Research Programme 2024 (VAN 2024), Ministry of Education, and the project of Masaryk University 'MUNI/A/1648/2024'. The NILU-UV measurements at Troll/Trollhaugen have been financed by the Norwegian Ministry of Climate and Environment. Technical personnel from the Norwegian Polar Institute are responsible for the daily maintenance of the instrument.

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
