# Peer review of "Evaluation of factors affecting total ozone column and its trend at three Antarctic stations in the years 2007–2023"

_EGUsphere, 2025_

## Referee Comment (RC1)

In this study, the authors present the trends of total column ozone at three Antarctic ozone monitoring stations over the period 2007-2023, both annually and during the spring months. They also discuss the impact of several dynamical processes on Total Column Ozone variability at the three monitoring stations and over the Antarctic more broadly using the MERRA-2 reanalysis. They present trends derived using the LOTUS (Long-term Ozone Trends and Uncertainties in the Stratosphere model) for the first time on ground-based and satellite overpass datasets in Antarctica. Finally, the authors offer a detailed analysis of the TCO variability during the 2019 and 2020 ozone holes. I think this is a very wonderful and timely study, as complications surrounding the identification of ozone hole recovery in Antarctica due to dynamic variability are still being discovered and actively discussed. This study fits ACP very well, and this reviewer recommends publication after revisions.

General comment about the logical structure and flow of the paper:

The results section is a little hard to follow because it is organized by station (i.e., 4.2 Marambio, 4.3 Troll, 4.4 Concordia), but there are discussions in some sections that contain results from all three stations. For example, the statistical tests (Durbin-Watson test) on lines 219-22, the discussion of the standard coefficients in lines 236-241 and again in 291-298. An alternative suggestion is to organize this section not by station, but by topic: trends, statistical tests, and then proxy regression coefficients.

Along the same lines, including a single table for the trends would enable an easy comparison. The overall trend estimate is a major result of this paper, and it was difficult to find and compare at first glance. One single table reporting the trend, uncertainty, p-value, and adjusted R^2 would be very helpful.

**General comment about the conclusion:**

The conclusion lacks a discussion about the broader implications and interpretation of the trends: what does it mean for monitoring for the ozone hole recovery identification and uncertainty? It is a good and interesting result, and the authors should discuss how it will impact the larger discussion of ozone hole identification. How should the reader interpret the trends? For example, the LOUTS model (and other MLR models) is often used as a method to ascribe dynamical and natural TCO variability so that the resulting trend can be interpreted as ozone-depleting gas (ODS) chemistry-related. Is that the goal in this study, and how should the reported trend be interpreted by the reader?

**General comment about limitations.**

The authors could expand upon the limitations in this study and how they may impact the interpretation. First, what is the uncertainty of the results and conclusions due to the temperature and cross sections used in the instruments, and from the merging of the satellites with the ground-based instrumentation? How does this impact the seasonal trend? Second, what is the impact of not using the LOUTS model on the monthly spring trends (September, October, November)? The authors are recommended to discuss how 'simple' linear trends do or do not compare with studies that use dynamical proxies in their trend analysis. Finally, the authors start the regression analysis in 2007, which is different from the LOTUS (SPARC, etc) report and most other LOTUS studies, which report the post-2000 trend. When comparing this study's results to others, this is an important caveat.

**Specific Comments:**

Line 50, Johnson is spelled incorrectly in the citation.

Line 58: Since this is the first introduction to the LOUTS model in the paper, I would recommend moving the citation (SPARC/LOTUS 2019) to here.

Line 88-90: The authors could also include information about the ozone cross sections and effective temperatures for the other instruments in addition to the SAOZ instrument. I would also recommend a short discussion of the impact of the chosen cross-section in the TCO retrieval on the trends, especially for Antarctic springtime ozone trends. Please see Fragkos et. al. 2013 and Rodeondas et.al 2014. Redondas et al. (2014) point out that the Brewers (as of the writing of their paper), in particular, do not use a correction factor for the effective temperature where SAOZ does. If the authors think it appropriate, please discuss the impact the cross-section and assumed effective temperatures may or may not have on the trends, particularly the seasonal (springtime) trends.

Line 135, Did you mean 25 missing measurements (like in the Bernet paper)? Or was there a reason not to include months with more than 5 missing measurements?

Line 142: Please include that the primary objective of LOTUS in the first phase was to derive the trends using global satellite datasets. Now in the third phase, there is a focus on trends derived from individual monitoring stations, regional variability, and representation of dynamic and physical processes.

Line 160: Please discuss how seasons with missing data and merging satellite with ground-based instruments could bias the harmonic fitting in the LOTUS model.

Line 164-170: This section has some repetitive information about the construction of the QBO proxies. It repeats some of the same wording from the other studies (particularly the Bernet 2023 paper). I would suggest making this section more concise and citing the other papers instead of repeating their discussions here.

Line 172: Can you clarify the statement that there are no statistically significant trends in the predictors? Your caption for figure S1 states: "Time series of predictors supplemented with a linear trend (red; except for the solar and QBO predictors), which was statistically insignificant at the significance level of p = 0.05.") This leads this reviewer to believe there are significant trends in all the predictors but the solar and QBO. This reviewer expected that there should be a trend in at least some proxies. Also, please clarify if the predictors were or were not detrended prior to the regression? It will have an impact on the interpretation of the trends.

Line 180: One suggestion, if you have time (or maybe this is for a separate paper), is to look at the trends in just the overpass dataset. The comparison between the trends derived from the overpass dataset alone and the trends derived from the ground-based supplemented with the overpass dataset would give an indication of the uncertainty in the trend due to any inhomogeneities between the GB and overpass datasets (up to 9% for some months in your own analysis).

Line 218-222: Since this section discusses all three stations, perhaps it's best to move it from the Marambio station section. Alternatively, move the discussions of each individual station to their respective sections. However, I would suggest the former.

Lines 287-298: Please consider clarifying this section. The authors appear to be comparing their results to the results of Bernet (2023). However, it is a little confusing which parts were discussed by Bernet (2023) and which were the results from this study. It seems to be line 291, but please clarify.

Line 359: Johnson is spelled incorrectly

Line 431; NOAA PSL, 2025, in the citation for the NASA OMI link should be removed.

| Line 437: Please give the link to the LOTUS model code here: https://usask-arg.github.io/lotus-regression/ |
|------------------------------------------------------------------------------------------------------------|
| regression/                                                                                                |
|                                                                                                            |
|                                                                                                            |
|                                                                                                            |
|                                                                                                            |
|                                                                                                            |
|                                                                                                            |
|                                                                                                            |
|                                                                                                            |
|                                                                                                            |
|                                                                                                            |
|                                                                                                            |
|                                                                                                            |
|                                                                                                            |
|                                                                                                            |
|                                                                                                            |
|                                                                                                            |
|                                                                                                            |

---

## Author Comment (AC1)

**Reply to Anonymous Referee #1 review of the manuscript acp-2025-3963**

**Evaluation of factors affecting TOC and its trend at three Antarctic stations in the years 2007–2023**

**David Tichopád on behalf of all co-authors**

**We sincerely thank Anonymous Referee #1 for the time dedicated to reviewing our work and for the constructive feedback. Your valuable comments have significantly contributed to the improvement of our manuscript.**

**Please find our answers below (in red)**

In this study, the authors present the trends of total column ozone at three Antarctic ozone monitoring stations over the period 2007-2023, both annually and during the spring months. They also discuss the impact of several dynamical processes on Total Column Ozone variability at the three monitoring stations and over the Antarctic more broadly using the MERRA-2 reanalysis. They present trends derived using the LOTUS (Long-term Ozone Trends and Uncertainties in the Stratosphere model) for the first time on ground-based and satellite overpass datasets in Antarctica. Finally, the authors offer a detailed analysis of the TCO variability during the 2019 and 2020 ozone holes. I think this is a very wonderful and timely study, as complications surrounding the identification of ozone hole recovery in Antarctica due to dynamic variability are still being discovered and actively discussed. This study fits ACP very well, and this reviewer recommends publication after revisions.

We sincerely thank the reviewer for their positive and encouraging assessment of our work. We are glad that the study is considered timely and relevant, and we appreciate the acknowledgement of the importance of addressing the complexities associated with identifying ozone hole recovery in Antarctica.

General comment about the logical structure and flow of the paper:

The results section is a little hard to follow because it is organized by station (i.e., 4.2 Marambio, 4.3 Troll, 4.4 Concordia), but there are discussions in some sections that contain results from all three stations. For example, the statistical tests (Durbin-Watson test) on lines 219-22, the discussion of the standard coefficients in lines 236-241 and again in 291-298. An alternative suggestion is to organize this section not by station, but by topic: trends, statistical tests, and then proxy regression coefficients.

We thank the reviewer for this constructive suggestion. We agree that organizing the results section by topic rather than by station can improve clarity and readability. In the revised manuscript, we have restructured the section to present the results under the headings "Trends," "Statistical Tests," and "Proxy Regression Coefficients." We believe this new organization allows for a more coherent comparison across all three stations and makes the key findings easier to follow.

Along the same lines, including a single table for the trends would enable an easy comparison. The overall trend estimate is a major result of this paper, and it was difficult to find and compare at first glance. One single table reporting the trend, uncertainty, p-value, and adjusted R^2 would be very helpful.

We agree that a single table summarising the trends, uncertainties, p-values, and adjusted $R^2$ would greatly improve clarity and allow for easier comparison (Tab. R1). A new table has been added to the Results section in the revised manuscript to present these key results.

*Tab. R1 Linear trends of TOC at the three Antarctic stations (Marambio, Troll, and Concordia) in 2007–2023. The table presents the estimated trend (DU/decade), the associated uncertainty, the p-value, and the adjusted $R^2$ for each station. A statistically significant trend is marked in bold (p < 0.05)*

| Station | Trend [DU/decade] | Uncertainty [DU/decade] | p-value | adjusted $R^2$ |
|---|---|---|---|---|
| Marambio | **3.43** | ±3.22 | 0.04 | 0.94 |
| Troll | -1.09 | ±3.91 | 0.58 | 0.97 |
| Concordia | 1.15 | ±4.25 | 0.59 | 0.95 |

General comment about the conclusion:

The conclusion lacks a discussion about the broader implications and interpretation of the trends: what does it mean for monitoring for the ozone hole recovery identification and uncertainty? It is a good and interesting result, and the authors should discuss how it will impact the larger discussion of ozone hole identification. How should the reader interpret the trends? For example, the LOUTS model (and other MLR models) is often used as a method to ascribe dynamical and natural TCO variability so that the resulting trend can be interpreted as ozone-depleting gas (ODS) chemistry-related. Is that the goal in this study, and how should the reported trend be interpreted by the reader?

We agree that discussing the broader implications of the trends is essential. In the revised manuscript, we have expanded the Conclusion section to clarify how the reported trends can be interpreted in the context of ozone hole recovery and associated uncertainties.

General comment about limitations.

The authors could expand upon the limitations in this study and how they may impact the interpretation. First, what is the uncertainty of the results and conclusions due to the temperature and cross sections used in the instruments, and from the merging of the satellites with the ground-based instrumentation? How does this impact the seasonal trend? Second, what is the impact of not using the LOUTS model on the monthly spring trends (September, October, November)? The authors are recommended to discuss how 'simple' linear trends do or do not compare with studies that use dynamical proxies in their trend analysis. Finally, the authors start the regression analysis in 2007, which is different from the LOTUS (SPARC, etc) report and most other LOTUS studies, which report the post-2000 trend. When comparing this study's results to others, this is an important caveat.

Thank you for this comment, and we will respond point by point:

1) We calculated the LOTUS regression (Tab. R2) for, in addition to the compiled time-series (used in the study), separately for the OMI data and the MERRA-2 data (the same months were used). In the case of the Marambio station, the trend is statistically significant for all datasets, but the magnitudes of the trend are slightly different. Also, in the case of the other stations, where the trends are statistically insignificant in all cases, the magnitudes of the trends differ slightly. These differences can be caused by the bias between ground measurements and the OMI and MERRA-2 data, which in mean reaches 1-3%. However, it reaches values up to 10 % within months (most at the Concordia station).

*Tab. R2 Linear trends of TOC at the three Antarctic stations (Marambio, Troll, and Concordia) in 2007–2023 compiled time series, OMI and MERRA-2 data. The table presents the estimated trend (DU/decade), the associated uncertainty, the p-value, and the adjusted $R^2$ for each station. A statistically significant trend is marked in bold (p < 0.05).*

| Station | Fit results | Trend [DU/decade] | Uncertainty [DU/decade] | p-value | Adjusted $R^2$ |
|---|---|---|---|---|---|
| Marambio | Compiled | **3.43** | ±3.22 | 0.04 | 0.94 |
| | OMI | **4.58** | ±2.97 | 0.00 | 0.95 |
| | MERRA-2 | **4.20** | ±3.00 | 0.01 | 0.95 |
| Troll | Compiled | -1.09 | ±3.91 | 0.58 | 0.97 |
| | OMI | 2.42 | ±3.30 | 0.15 | 0.98 |
| | MERRA-2 | 1.59 | ±3.31 | 0.34 | 0.98 |
| Concordia | Compiled | 1.15 | ±4.25 | 0.59 | 0.95 |
| | OMI | 1.47 | ±4.61 | 0.53 | 0.95 |
| | MERRA-2 | 0.47 | ±4.40 | 0.83 | 0.95 |

The uncertainty related to the temperature dependence of absorption coefficients and the choice of cross-section datasets for Brewer instruments is small, typically below about 1 %. The temperature effect is <0.01 % K$^{-1}$ (Redondas et al., 2014), resulting in a negligible impact on the derived DS ozone trends.

For cloud-free conditions and SZA < 70 degrees the overall uncertainty in NILU-UV is estimated to be ±5% (Sztipanov et al., 2020). For SZA > 70 degrees the impact of cloudiness, the vertical profile of ozone and temperature, the imperfect cosine response of the instrument, and the absolute calibration error will reduce the accuracy of the TOC values (Sztipanov et al., 2020; Kazantzidis et al., 2009).

The SAOZ instrument uses Bogumil et al. (2003) ozone cross sections in the visible, where temperature dependency is practically negligible. The systematic uncertainties on the ozone absorption cross sections, considering slight dependence on temperature, are approximately 3% in the SAOZ spectral range (Orphal, 2003). A complete budget estimation can be found in Hendrick et al. (2011).

This limitation is, however, now noted in the manuscript.

2) The LOTUS regression could not be applied for the monthly spring trends (September, October, November) because the variance inflation factor (VIF) exceeded 10, indicating strong multicollinearity among the predictors. As a result, assessing individual predictor contributions using LOTUS was not feasible for these months.

3) The comparison of our linear trends with other studies using dynamical proxies is addressed by comparing the TOC trends derived from our analysis with those obtained from OMI overpass and MERRA-2 datasets using the LOTUS regression. The results of this comparison are presented in a table in the Supplement, providing context for how our post-2007 trends relate to other studies that report post-2000 trends.

Specific Comments:

Line 50, Johnson is spelled incorrectly in the citation.

Corrected

Line 58: Since this is the first introduction to the LOUTS model in the paper, I would recommend moving the citation (SPARC/LOTUS 2019) to here.

Corrected

Line 88-90: The authors could also include information about the ozone cross sections and effective temperatures for the other instruments in addition to the SAOZ instrument. I would also recommend a short discussion of the impact of the chosen cross-section in the TCO retrieval on the trends, especially for Antarctic springtime ozone trends. Please see Fragkos et. al. 2013 and Rodeondas et.al 2014. Redondas et al. (2014) point out that the Brewers (as of the writing of their paper), in particular, do not use a correction factor for the effective temperature where SAOZ does. If the authors think it appropriate, please discuss the impact the cross section and assumed effective temperatures may or may not have on the trends, particularly the seasonal (springtime) trends.

Thank you, as we said earlier in the general comment: The uncertainty related to the temperature dependence of absorption coefficients and the choice of cross-section datasets for Brewer instruments is small, typically below about 1 %. The temperature effect is <0.01 % $K^{-1}$ (Redondas et al., 2014), resulting in a negligible impact on the derived DS ozone trends. This limitation is now noted in the manuscript.

Line 135, Did you mean 25 missing measurements (like in the Bernet paper)? Or was there a reason not to include months with more than 5 missing measurements?

We used a threshold of up to 5 missing daily measurements per month to ensure representative monthly averages. In contrast, Bernet et al. (2023) excluded months with fewer than 25 measurement days, which is conceptually similar. This choice was made to balance data coverage and representativity; months exceeding the threshold were excluded (April to August for all three stations). A clarification has been added to the manuscript.

Line 142: Please include that the primary objective of LOTUS in the first phase was to derive the trends using global satellite datasets. Now in the third phase, there is a focus on trends

derived from individual monitoring stations, regional variability, and representation of dynamic and physical processes.

Thank you, included.

Line 160: Please discuss how seasons with missing data and merging satellite with ground based instruments could bias the harmonic fitting in the LOTUS model.

Line 164-170: This section has some repetitive information about the construction of the QBO proxies. It repeats some of the same wording from the other studies (particularly the Bernet 2023 paper). I would suggest making this section more concise and citing the other papers instead of repeating their discussions here.

Corrected

Line 172: Can you clarify the statement that there are no statistically significant trends in the predictors? Your caption for figure S1 states: "Time series of predictors supplemented with a linear trend (red; except for the solar and QBO predictors), which was statistically insignificant at the significance level of p = 0.05.") This leads this reviewer to believe there are significant trends in all the predictors but the solar and QBO. This reviewer expected that there should be a trend in at least some proxies. Also, please clarify if the predictors were or were not detrended prior to the regression? It will have an impact on the interpretation of the trends.

The predictors were not detrended prior to regression because linear trends were not statistically significant in any of the predictors (p > 0.05). The exception is the solar cycle (Solar), which exhibits an ~11-year periodic cycle that is not linear, and removing it would eliminate any physically relevant signal. Therefore, detrending was not methodologically justified and could have biased the interpretation of the results.

Line 180: One suggestion, if you have time (or maybe this is for a separate paper), is to look at the trends in just the overpass dataset. The comparison between the trends derived from the overpass dataset alone and the trends derived from the ground-based supplemented with the overpass dataset would give an indication of the uncertainty in the trend due to any inhomogeneities between the GB and overpass datasets (up to 9% for some months in your own analysis).

We thank the reviewer for this suggestion. We are currently planning to prepare a separate manuscript, in which we will compare trends derived from the overpass dataset alone with those derived from the ground-based dataset supplemented with the overpass data.

Line 218-222: Since this section discusses all three stations, perhaps it's best to move it from the Marambio station section. Alternatively, move the discussions of each individual station to their respective sections. However, I would suggest the former.

We thank the reviewer for this suggestion. This issue has already been addressed based on our previous general comment regarding the logical structure and flow of the manuscript.

Lines 287-298: Please consider clarifying this section. The authors appear to be comparing their results to the results of Bernet (2023). However, it is a little confusing which parts were discussed by Bernet (2023) and which were the results from this study. It seems to be line 291, but please clarify.

Thank you, corrected.

Line 359: Johnson is spelled incorrectly

Corrected

Line 431; NOAA PSL, 2025, in the citation for the NASA OMI link should be removed.

Corrected

Line 437: Please give the link to the LOTUS model code here: https://usask-arg.github.io/lotus regression

The link to the LOTUS model code in R will be added to the manuscript during the revision process.

References:

Bernet, L., Svendby, T., Hansen, G., Orsolini, Y., Dahlback, A., Goutail, F., Pazmiño, A., Petkov, B., and Kylling, A.: Total ozone trends at three northern high-latitude stations, Atmos Chem Phys, 23, 4165–4184, https://doi.org/10.5194/acp-23-4165-2023, 2023.

Bogumil, K., Orphal, J., Homann, T., Voigt, S., Spietz, P., Fleischmann, O. C., Vogel, A., Hartmann, M., Bovensmann, H., Frerik, J., and Burrows, J. P.: Measurements of molecular absorption spectra with the SCIAMACHY Pre-Flight Model: Instrument characterization and reference spectra for atmospheric remote sensing in the 230-2380 nm region, J. Photochem. Photobiol. A, 157, 167–184, 2003.

Hendrick, F., Pommereau, J.-P., Goutail, F., Evans, R. D., Ionov, D., Pazmino, A., Kyrö, E., Held, G., Eriksen, P., Dorokhov, V., Gil, M., and Van Roozendael, M.: NDACC/SAOZ UV-visible total ozone measurements: improved retrieval and comparison with correlative ground-based and satellite observations, Atmos. Chem. Phys., 11, 5975–5995, https://doi.org/10.5194/acp-11-5975-2011, 2011.

Kazantzidis, A. Bais, A. F., Zempila, M. M., Meleti, C., Eleftheratos, K., Zerefos, C. S.: "Evaluation of ozone column measurements over Greece with NILU-UV multi-channel radiometers," Int. J. Remote. Sens. 30, 4273-4281, 2009.

Orphal, J.: A critical review of the absorption cross-sections of O3 and NO2 in the 240-790 nm region, J. Photochem. Photobiol. A: Chemistry, 157, 185–209, 2003.

Redondas, A., Evans, R., Stuebi, R., Köhler, U., and Weber, M.: Evaluation of the use of five laboratory-determined ozone absorption cross sections in Brewer and Dobson retrieval algorithms, Atmos. Chem. Phys., 14, 1635–1648, https://doi.org/10.5194/acp-14-1635-2014, 2014.

Sztipanov, M., Tumeh, L., Li, W., Svendby, T., Kylling, A., Dahlback, A., Stamnes, J., Hansen, G.H., Stamnes, K.: Ground-based measurements of total ozone column amount with a multichannel moderate-bandwidth filter instrument at the Troll research station, Antarctica, Appl. Opt., 59, 97-106, doi: 10.1364/AO.59.000097, 2020.

---

## Author Comment (AC2)

**Reply to Anonymous Referee #2 review of the manuscript acp-2025-3963**

**Evaluation of factors affecting TOC and its trend at three Antarctic stations in the years 2007–2023**

**David Tichopád on behalf of all co-authors**

We sincerely thank Anonymous Referee #2 for the time dedicated to reviewing our work and for the constructive feedback. Your valuable comments have significantly contributed to the improvement of our manuscript.

Please find our answers below (in red)

General Comments:

This manuscript describes a long-term time series analysis of ground-based records from three stations in Antarctica. The authors use the multiple regression model developed by the LOTUS group, with additional parameters optimized for variability at polar latitudes. The authors additionally analyzed MERRA-2 reanalysis output to assess the spatial distribution of the various parameters. The study is very relevant and the manuscript clearly written and well-referenced. The figures, tables and supplemental material are clearly presented. I recommend publication after the following issues are addressed.

We thank the reviewer for their careful reading of our manuscript and for the constructive comments. We appreciate the positive assessment of the study and have addressed all points raised in the revised manuscript.

Specific Comments:

L68: The sentence starting "While the B199 instrument offers very high accuracy…" is somewhat confusing. I believe the 0.15% from Scarnato et al. refers to the precision rather than the accuracy and is for the double Brewer instrument in general. Also, the wording should clarify that the direct sun measurements are the most precise. Something like: "To assure the highest precision, only direct sun measurements were utilized."

Corrected

L99: Including a different data set, particularly at the endpoints, can have a notable impact on the trend even if the offsets are small. It will be difficult to compare with other studies because of the specific period of the fit, starting later in the recovery time period, in 2007. To address this, I think it would be instructive, either in the paper or in the supplemental material, to show a figure similar to Figure 5 but use the OMI overpass and MERRA-2 overpass data at each station (time series shown in Figure 2) to get an idea of the possible range of values from different data sources. There are notable differences in the time series as shown in Figure 2, including a small drift at Troll and Concordia.

Thank you for highlighting the sensitivity of the trend estimates to the choice of dataset and to the length of the fitting period. We agree that differences at the endpoints, even if small, may influence the derived trends and complicate comparisons with other studies. In response to this comment, we have added supplementary figures analogous to Figure 5 (Fig. R1 and R2), in which the trends are calculated using OMI overpass data and MERRA-2 overpass data at each station (based on the time series shown in Figure 2).

[Figure]

*Fig. R1 Predictor contributions to the annual regression fit at Marambio (a), Troll (b) and Concordia (c) for OMI overpass data. Standardised coefficients indicate the percentage change in TOC associated with a one standard deviation change in the predictor. Light blue bars denote predictors whose effect on ozone is not statistically significant (p-value of the coefficient <0.05).*

When comparing all three figures (the compiled time series, the OMI overpass data, and the MERRA-2 data), small differences are apparent. At Marambio, the influence of T100 remains statistically significant, but QBOd becomes statistically insignificant in both the OMI and MERRA-2 datasets, while the solar factor becomes statistically significant. At Troll, the results are similar, except that the QBOc predictor is no longer statistically significant when using OMI and MERRA-2 data. At Concordia, the solar factor is not statistically significant when using either the OMI or MERRA-2 datasets.

[Figure]

*Fig. R2 Predictor contributions to the annual regression fit at Marambio (a), Troll (b) and Concordia (c) for MERRA-2 data. Standardised coefficients indicate the percentage change in TOC associated with a one standard deviation change in the predictor. Light blue bars denote predictors whose effect on ozone is not statistically significant (p-value of the coefficient <0.05).*

L235: Here and throughout the manuscript when discussing the QBO fits, can the authors explain the relevance of the individual QBO terms. I understand why multiple EOF principal component time series are used, but I do not believe these terms can be physically interpreted individually. For example in Figure 4, rather than show the individual terms, I believe this result is more easily understood if the terms are re-added to represent the net QBO variability. I would also be interested to know if the net QBO fit was statistically significant at each station. I believe the authors can apply a joint F-test to determine this. Also in Figure 10, a panel can be added that shows the reconstruction of the net QBO for the two years. I realize the figures and discussion are set up to address the individual terms, I would just like to see the examples of the full QBO signal expressed in DU in Figures 4 and 10, and if possible an estimate of the significance of the full QBO signal.

Thank you for this very valuable comment. In response, we have added a curve representing the sum of the four QBO components in Figure 4, as well as in Figures S4 and S5. We have also assessed the statistical significance of the net QBO fit using a joint F-test. The results show that the combined QBO signal is statistically significant only at Troll (F = 4.51, p = 0.002), while for the other stations it is not statistically significant. These results have been included in the revised manuscript.

We have also added the net QBO signal to Figure 10 and performed a joint F-test for each grid point (Fig. R3), which will be included in the supplementary material and referenced in the manuscript.

[Figure]

*Figure R3 Joint F-test for the net QBO signal for each grid point. Colours indicate the F-statistic values. The unshaded area is statistically significant (p<0.05).*

Line 245: This is a nice figure, can the trend term be added as well? It is a little out of place because it covers all three stations, it might fit better after the regression fits for each station are presented, but this change is not mandatory. The same results using the OMI and MERRA-2 overpass time series would be very interesting as mentioned before, this plot could be part of the supplemental information but referred to in the text. Such a plot would also make it easier for the reader to compare the results in Figure 9 to the station results.

We thank the reviewer for this helpful suggestion. As noted in our response to the L99, the comment corresponding plots based on the OMI and MERRA-2 overpass datasets have been added to the supplemental material. We have also calculated the individual trends using the LOTUS regression and present these results in a supplementary table (Tab. R1). In addition, we have reorganized the section containing this figure to improve clarity and readability.

*Table R1 Linear trends of TOC at the three Antarctic stations (Marambio, Troll, and Concordia) in 2007–2023 compiled time series, OMI and MERRA-2 data. The table presents the estimated trend (DU/decade), the associated uncertainty, the p-value, and the adjusted $R^2$ for each station. A statistically significant trend is marked in bold ($p < 0.05$).*

| Station | Fit results | Trend [DU/decade] | Uncertainty [DU/decade] | p-value | Adjusted $R^2$ |
|---------|-------------|-------------------|-------------------------|---------|----------------|
| Marambio | Compiled | **3.43** | ±3.22 | 0.04 | 0.94 |
| | OMI | **4.58** | ±2.97 | 0.00 | 0.95 |
| | MERRA-2 | **4.20** | ±3.00 | 0.01 | 0.95 |
| Troll | Compiled | -1.09 | ±3.91 | 0.58 | 0.97 |
| | OMI | 2.42 | ±3.30 | 0.15 | 0.98 |
| | MERRA-2 | 1.59 | ±3.31 | 0.34 | 0.98 |
| Concordia | Compiled | 1.15 | ±4.25 | 0.59 | 0.95 |
| | OMI | 1.47 | ±4.61 | 0.53 | 0.95 |
| | MERRA-2 | 0.47 | ±4.40 | 0.83 | 0.95 |

Line 392: I would caution the authors not to assign too much causality to some of the fits. For example, when comparing 2019 and 2020, the QBOa and QBOd terms switch signs, as do the ENSO and IOD terms, but this is due to the proxy signals changing sign over this period, not due to the conditions of a warmer or colder polar stratosphere (see Figure S1). The temperature and EHF are related to the polar dynamics, but the QBO/ENSO/IOD terms vary according to the time scale of those forcings and just happen to change sign from 2019 to 2020. It is possible that the QBO phase impacts the wave activity and thus the vortex, but to show this the authors would need to assess the QBO phase over a series of cold and warm polar conditions. The text reads as though a warm vortex produces a QBO signal that is one sign, and a cold vortex a QBO signal of the other sign. This may not be the intention, but it should be carefully worded to avoid inaccurate (or at least unproved) associations.

We agree that the sign changes in the QBO, ENSO, and IOD terms between 2019 and 2020 are driven by the behaviour of the proxy time series themselves, and should not be interpreted as a direct causal response to warmer or colder polar vortex conditions. Our intention was not to imply such a causal relationship, and we have revised the text accordingly to avoid any unintended interpretation.

Conclusions: It would be interesting to see comparisons of the derived trends with trends from other studies. This might be difficult because of the variable time periods between studies. But again, a comparison with the satellite and reanalysis overpass time series would be useful in place of outside comparisons.

We thank the reviewer for this suggestion. The derived trends are compared with OMI overpass data and MERRA-2 reanalysis in Table X (see comment at L245). This comparison provides context and allows assessment of the consistency of our results with these datasets.

Technical Corrections:

Title: suggest spelling out TOC in the title

Corrected

L19: any time the trend is given, the uncertainty estimate should be included.

Corrected

L43: suggest wording change: "a strong wave-1 disturbance developed"

Corrected

L50 Jonson -> Johnson

Corrected

L116: SBUV can be removed here, it is not included in the description and SBUV is not relevant to MERRA-2 after October 2004.

Corrected

L156: please clarify in the text, the equatorial zonal mean wind at seven pressure levels between 70-10 hPa were used

Corrected

L189: the lowest mean deviation

Corrected

L212: suggest removing "and the lowest in September-October and January -April" I think it is sufficient to say the largest variability is in November-December.

Corrected

L230: Can the authors say more here about whether the ENSO results agree with the study by Lin and Qian (2019).

Corrected

We have decided to remove this sentence, as a direct comparison with Lin and Qian (2019) is difficult because their study focuses on the spatial distribution of ozone.

L235: suggest adding tick marks for each year to make the plot easier to read.

Tick marks for each year are already included in the plot, which should facilitate reading the time series.

L263: Is this because Marambio is sometimes in the collar region?

It is indeed possible that the observed behaviour at Marambio is influenced by its occasional location within the collar region.

L284: The last sentence in this paragraph is largely repetitive. I suggest removing it or revising the last sentence of the previous paragraph to include this information.

Corrected

L311-312: The Marambio trend in November is positive, and the trend for September is increasing but also not statistically significant. Suggest: "Interestingly, the trends for October and November and decreasing but not statistically significant at all stations except Marambio in November, while the trends for September at all stations are increasing, but also not statistically significant."

Corrected

L320: suggest adding tick marks for each year to make the plot easier to read. Also in Figures S1 and S4.

Tick marks for each year are already included in the plots, including Figures S1 and S4, to facilitate reading the time series.

L328-329: suggest simplifying the text here, possibly "Time series analysis at each grid point shows the spatial distribution of the fits to each parameter, which are expressed using standardized coefficients of determination."

Corrected

L338: suggest Lin and Qian (2019) shows that …

Corrected

L346: the equatorial zonal wind at seven pressure levels between 10 and 70 hPa.

Corrected

Line 359: Jonson -> Johnson

Corrected

Line 359 and 361: can remove "Studies by" and "A study of" to simplify the text.

Corrected

L376: suggest wording clarification: "slowdown the heterogenous reactions that activate Cl on the surface of PSCs, thus slowing ozone depletion and suppressing the formation… "

Corrected

Line 380: Was the further decomposition of the EHF done in the Shen et al., 2020 study or in this study? If in the Sten et al study I suggest "Based on further decomposition of the EHF, Shen et al. (2020) found… " to make it clear this was not part of the current work.

Corrected

L413: include trend uncertainty value in text

Corrected

---

## Referee Report (RR1)

Title: Evaluation of factors affecting TOC and its trend at three Antarctic stations in the years 2007–2023
Author(s): David Tichopád et al.
MS type: Research article
Submission ID: egusphere-2025-3963
Iteration: Revised submission

In this study, the authors present the annual and springtime trends of total column ozone at three Antarctic ozone monitoring stations (Marambio, Troll, and Concordia) over the period 2007-2023. They also discuss the impact of several dynamical processes on Total Column Ozone variability at the three monitoring stations and over the Antarctic more broadly, using the MERRA-2 reanalysis. They present trends derived using the LOTUS (Long-term Ozone Trends and Uncertainties in the Stratosphere) multi-linear regression model for the first time on ground-based and satellite overpass datasets in Antarctica. They found an increasing trend, significant only at the Marambio station, which is at the polar vortex edge. Troll and Concordia in the interior of the continent were not found to have a trend statistically different from zero. From the results of the coefficients of the explanatory variables in the model (which served as proxies for dynamical and natural variability), they found that the lower stratosphere (100mb) temperature had the largest influence on total column ozone variability, with the Qusai-Biennial oscillation and the eddy heat flux (as a proxy for the Brewer-Dobson Circulation) also had statistically significant influence at the stations.

The trend estimates derived using the same model, but with datasets from the Ozone Monitoring Instrument (OMI) satellite overpass and the MERRA-2 reanalysis, were consistent with the ground-based trend estimates, with minor deviations. The LOTUS model was also run on each MERRA-2 grid point over the Antarctic continent to derive the spatial distribution of the factors influencing total column ozone variability. They found that the temperature in the lower stratosphere is the strongest influence on the entire content; eddy heat flux is mostly significant; and the El Niño/Southern Oscillation and Indian Ocean Dipole were found to have insignificant (at the 95% confidence level) influence on total column ozone.

Finally, the authors offer a detailed analysis of TCO variability during the 2019 and 2020 ozone holes, finding that the exceptional sudden stratospheric warming event in 2019 significantly increased total column ozone over Antarctica. They conclude that the ozone layer is thickening and recovering at the continent's edge, but polar vortex dynamics and declining stratospheric temperatures are complicating efforts to detect ozone recovery in the Antarctic interior. They rightfully highlight the need to continue long-term monitoring of stratospheric ozone.

I think this is a very wonderful and timely study, as complications surrounding the identification of ozone hole recovery in Antarctica due to dynamic variability are still being discovered and actively discussed, and it is perfectly appropriate for publication in ACP. The authors have addressed all of my concerns and questions in their revised manuscript, and I recommend publication as-is with this revised submission.

---

## Author Response (AR2)

**Reply to Anonymous Referee #1 review of the manuscript acp-2025-3963**

**Evaluation of factors affecting total ozone column and its trend at three Antarctic stations in the years 2007–2023**

**David Tichopád on behalf of all co-authors**

We sincerely thank Anonymous Referee #1 for the very positive and encouraging evaluation of our manuscript. We appreciate the careful reading and the supportive comments. As the reviewer indicates that all concerns have been satisfactorily addressed and recommends publication as is, no further changes were made to the manuscript.

**Reply to Anonymous Referee #2 review of the manuscript acp-2025-3963**

**Evaluation of factors affecting total ozone column and its trend at three Antarctic stations in the years 2007–2023**

**David Tichopád on behalf of all co-authors**

We very thank Anonymous Referee #2 for the positive evaluation of the revised manuscript and for appreciating the additional analyses included in this version. All comments have been addressed, and the manuscript has been revised accordingly. Detailed responses to each point are provided below.

**L131: OMI retrieves**

Corrected

**Line 194: suggest moving new sentence on net QBO signal down one sentence (Line 197) to keep the reasoning for using multiple QBO EOFs together.**

Corrected

**Line 242: Figure 3 -> Figure 3a**

Corrected

**Line 327: the wording here is a bit confusing. Suggest: According to the study by Klekociuk et al. (2015) the 2013 early winter temperatures were anomalously low in the polar stratosphere, with a concomitant strong and stable polar vortex supporting the potential for strong ozone depletion.**

Corrected

**Line 469: sections 4.2 -> section 4.2**

Corrected

**Line 473: with a statistically significant negative effect (to keep consistent with previous wording)**

Corrected

**Line 473: with a statistically significant negative effect (to keep consistent with previous wording)**

Corrected